

# Brief Communication: Measuring Rock Decelerations and Rotation Changes During Short Duration Ground Impacts

Andrin Caviezel[1] and Werner Gerber[2]

[1]WSL Institute for Snow and Avalanche Research SLF, CH-7260 Davos Dorf
[2]Swiss Federal Institute for Forest, Snow and Landscape Research WSL, CH-8903 Birmensdorf

**Correspondence:** Andrin Caviezel (andrin.caviezel@slf.ch)

**Abstract.** Rockfall trajectories are primarily influenced by ground contacts, causing changes in acceleration and rock rotation. The duration of contacts and its influence on the rock kinematics are highly variable and generally unknown. The lack of knowledge hinders the development and calibration of physics based rockfall trajectory models needed for hazard mitigation. To address this problem we placed three-axis gyroscopes and accelerometers in rocks of various sizes and shapes with the goal

of quantifying rock deceleration in natural terrain. Short ground contacts range between 8-15 milliseconds, longer contacts between 50-70 ms, totalling to only 6% of the runtime. Our results underscore the highly non-linear character of rock-ground interaction and the basic difficulties underlying rockfall hazard mitigation.

## 1   Introduction

A detailed understanding of object penetration into matter is essential both from a fundamental physics as well as geophysical

point of view. The relevant time scale spans form high-speed impacts of km/sec in planetary science to cm/s or mm/s in laboratory experiments of intruder sinking into granular beds. There exists a manifold on studies on penetration of objects into granular media or coefficients of restitution (see Altshuler et al. (2014); Asteriou et al. (2012) and references therein). On the other side, there is little understanding of processes arising from altered impact conditions, such as deviations from normal impact configurations, high rotational speeds of the impacting object, etc.. Consequently, the understanding of the mechanics of

rock-ground interactions poses a longstanding problem in rockfall engineering. This interaction defines the speed, jump height and dispersion of falling rocks in natural terrain. Because ground interaction controls rockfall runout distances and energy levels, it is the core problem when developing physics-based dynamic models for rockfall hazard mitigation and planning (Leine et al., 2014).

One approach to address the impact problematic is to use either dendrogeomorphic techniques to asses rockfall frequency and

distribution (Trappmann and Stoffel, 2015; Corona et al., 2017) and/or trajectory reconstruction via impact analysis (Paronuzzi, 2009; Saroglou et al., 2018). Hardly any data exists that directly measures rock-ground interactions during a rockfall event. A possible method to characterize ground impacts is to study surface scars left by falling rocks. For example, translational rock velocities can be determined by measuring the distance between ground contacts and the relative slope angle between the contact points. For an initial estimation it can be assumed that the jump height will be about 1/10 of the jump distance on





the slope. Based on such an assumption, a flight parabola is determined and the relevant velocities can be calculated (Gerber, 2015). If different jump heights are assumed, e.g. 1/8 or a 1/12, the maximum velocities shortly before ground impact will change by less than 10%. In many cases, this method suffices to obtain a rough estimate of the dissipative character of the ground interaction.

The problem with many approaches is that ground scaring is often difficult to physically interpret, especially if the rock is in a fast rotating, rolling motion. In this case the distances between ground contacts are extremely short and provide little information concerning the true velocity of the rock. Although the depth of the ground scar is an indication of the rebound mechanics at work, scar depths are highly variable, especially if the rock is "skipping" on the ground surface. Moreover, the analysis of rockfall traces provides little information of the mechanics of ground interaction, particularly if the relationship

between the translational and rotational kinematics of the rock are unknown.

    Newer studies in penetration studies make use of emerging microelectronical mechanical sensors to directly track the occurring motion (Sanchez-Colina et al., 2014; Gronz et al., 2016). Note, that to date, the major drawback of available multidegree of freedom inertial measurements units (IMU), is the range restriction to low accelerations (few tens of $g$). Because major application for such IMUs are unmanned aerial vehicle flight control, resistance to and measurement capabilities of heavy

impacts is not the main focus of chip makers.

    In this paper, we present novel and detailed in-situ measurements of high-impact ground interaction contact times, decelerations and changes in rock rotations using sensors inserted inside the rock. The resulting three-dimensional measurements yield detailed insights into how rocks behave, both in flight and upon contact with the ground. The measurements reveal the limits of field analyses, but also how experimental field campaigns can be constructed to obtain the necessary data needed to calibrate

constitutive relationships for dynamic rockfall models.

    However, before any conclusions can be reached or any further calculations made based on the results, the measurements must be subjected to a quality check and verified. At present we have little idea of the degree of acceleration reversal and change of rotational speed during impact, making it difficult to judge the accuracy of the measurements. Simple kinematic requirements must be fulfilled. For example, the acceleration measurements at rest must correspond to the value of gravitational acceleration

and indicate a value of zero in free flight. The purpose of this brief communication is therefore to elaborate on measurement frequencies and methods needed to capture the physical information required to study rock-ground interaction in natural terrain. We believe this information is necessary to develop better trajectory models for rockfall hazard mitigation.

## 2   METHODS OF MEASUREMENT AND EVALUATION

### 2.1   Field studies

The rockfall tests were performed in natural terrain. The test site, located near Tschamut in the Canton of Grisons, is a slope 50 m high with a maximum inclination of 42°, running down to a horizontal surface. The surface vegetation consists mainly of grass, with a few scattered shrubs in the upper, steeper part of the slope. The absence of tall vegetation and relatively smooth terrain allow a clear observational view and make the Tschamut site ideal for conducting rockfall experiments and filming



the rocks' movements. A release point at the top of the slope was selected, measured and used to release the rocks by simple dropping (no or little initial translational velocity and initial spin). The release point was selected to accommodate the transport of rocks, facilitating experimental data sets of more than 50 releases on a single day (i.e. with the same ground conditions including temperature and moisture content).

We present the results of one out of more than 50 trajectories captured in a test series specifically designed to investigate the role of rock shape on runout and dispersion, see Caviezel et al. (2018c, a). In this particular measured run, an artificially manufactured concrete block with an 0.3 m edge length and a mass of 44 kg was released. The symmetric and well-defined block shape was used as a control geometry in the rockfall experiments. The rock's corners and edges and were pared back a quarter to make the block less dice-shaped. A hole 68 mm in diameter was drilled through the block to accommodate the

sensor. The block's mass and volume (0.019 m$^3$) make it equivalent to a sphere with a radius of 0.165 m and a circumference of 1.04 m.

## 2.2   Sensor

In view of developments in consumer electronics for devices including tablets, mobile phones and unmanned aerial vehicles (UAVs), the measurement ranges and performances of available miniaturized motion sensors are steadily increasing. In-situ

data were recorded using a dedicated low-power sensor node, dubbed **StoneNode** (Figure 1a), see Caviezel et al. (2018b). The main components of **StoneNode** v1.0, which was used to record the data presented here, are a tri-axial accelerometer with a measurement range of 400 $g$ and an InvenSense 3-axis gyroscope recording up to 4'000 deg/s.

A micro-controller manufactured by Texas Instruments hosts the sensors and was selected for its low power consumption (roughly 3.6 mW at 3 V). Thus, an 1,100-mAh LiPo battery can gather 56 hours of data. Efficient data retrieval is ensured using

a plug-and-play USB device. For detailed information on the sensors used and comparison with other systems, see Niklaus et al. (2017) and Caviezel et al. (2018c, b).

## 2.3   Quality analysis

Before the measurements can be processed, the raw data need to be verified. Assuming that the sensors are functioning properly, the raw data should be checked for the following criteria:

– The measuring range of each individual sensor should not be exceeded.

   – When at rest, the rotational velocity should equal zero and the acceleration values should equal one, corresponding to gravitational acceleration.

   – During free fall, the rotational velocity should remain constant, with zero absolute acceleration; this analysis must be performed when there is relatively little rotation, because the influence of central acceleration will grow at higher rotational

velocities.

   – Theoretically, when rotational velocity is constant, the phenomenon of central acceleration should result in the measurement of higher values.



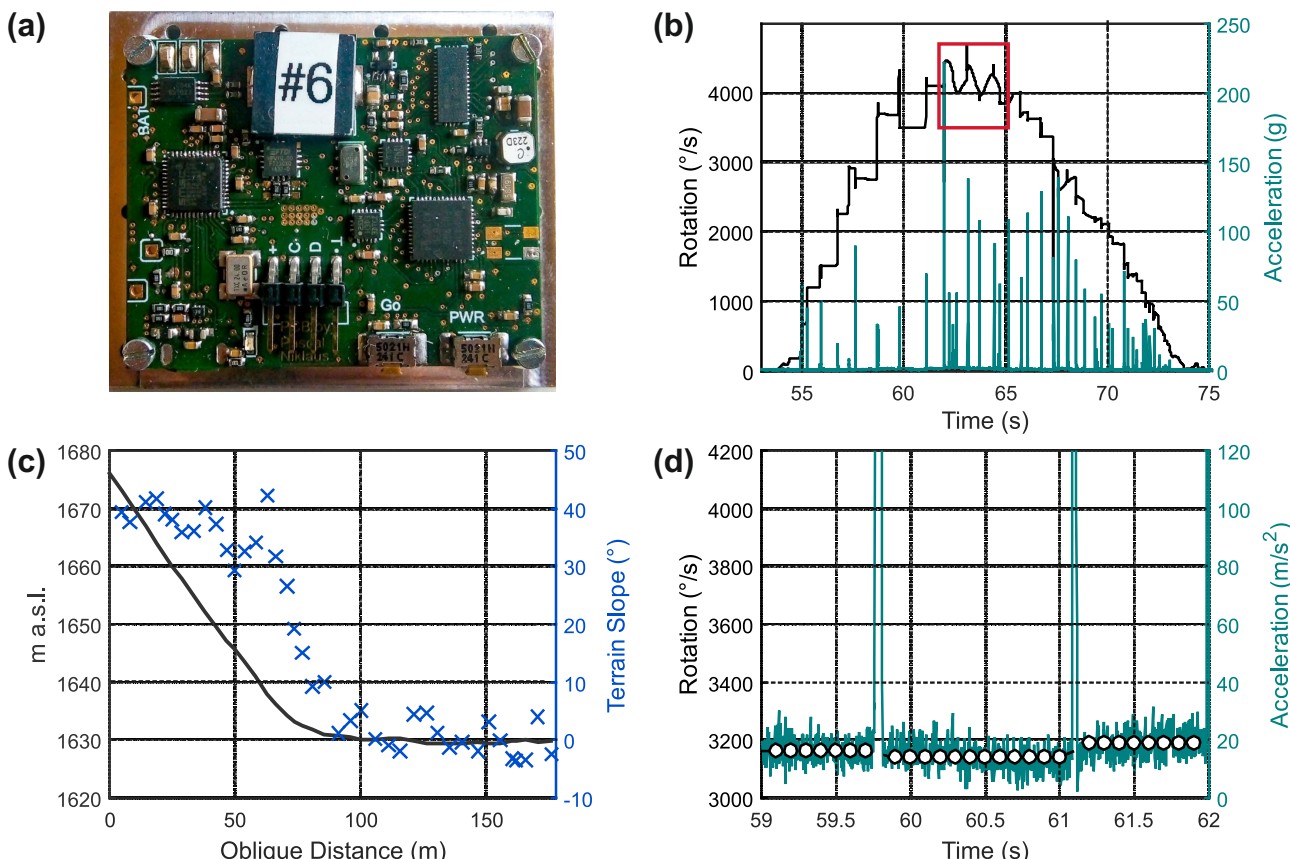

**Figure 1. (a)** An exposed micro-controller board hosting all the MEMS sensors, microSD card and a USB connector powered by an 1,100 mAh battery (both covered by the board). **(b)** Sensor data stream showing absolute rotational velocities and acceleration values during the 20-s movement phase (from 54 to 74 s), the rectangle indicating range with one saturated axis, **(c)** Slope distance of the projected trajectory of the stone with its location and slopes, **(d)** Absolute rotational velocities and acceleration values, plus mean acceleration values for calculating eccentricity.

The rotational velocities and acceleration values can then be correlated, representing the eccentrically fitted sensor as the rock's centre of gravity. In physical terms, this relationship can be expressed by the formula Eqn. 1:

$$R_e = \frac{a_Z}{\omega^2} \tag{1}$$

where $R_e$ is the sensor's eccentricity (m), $a_Z$ the central acceleration (m/s$^2$); $\omega$ angular velocity (rad/s). In theory, rotational differences should result in the same eccentricities. However, this only holds true for horizontal rotation: for vertical rotation, gravitational acceleration $g = 9.81$ m/s$^2$ should also be taken into account.



## 3 Results

### 3.1 General

The raw data comprise measurements starting from when the sensor was switched on until the block's deposition some 74 s later. The effective start of the rockfall occurred after around 54 s. During this period, 8,000 acceleration values for all three axes ($x$, $y$ and $z$) and 9,750 rotational values were measured. Analysis of frequency measurements yielded values of 400 Hz during acceleration and 487.5 Hz for rotation. During the 20-second rockfall from release to deposition, the block covered a horizontal distance of 147 m and negotiated a height difference of 49 m. The maximum inclination on site was -42°, dropping to zero and even +4° on the upslope of the depositional area. The effective fall trajectory's slope length was 162 m (Figure 1c).

Ground contacts are very clearly indicated by sharp peaks in the acceleration measurements and changes in rotational velocities. In steeper terrain, significantly fewer ground contacts were registered than in the runout zone. Whereas ground contacts occur every two seconds on steep slopes, on shallower inclines there are between twice and three times as many. Absolute rotation increases from an initial value of zero to 4'500 degrees per second (deg/s) before falling back to zero.

The maximum absolute acceleration value measured was 225 $g$ (at second 62.0). The rest of the measure values range below 140 $g$, and many were even less than 50 $g$ (Figure 1b).

## 4 Quality analysis procedure

The absolute rotational velocities and acceleration values were calculated and presented in the preceding section. The peak values of the individual measurements were checked. Due to the symmetry braking caused by the sensor hole, a main rotational axis exists which reaches the saturation limit between 62-64.6 s. This causes the resultant trace to predominantly feature the oscillating mode of the two remaining axis highlighted with the box in Fig. 1b.

The sensors are specified to high resolution capabilities of 0.122 deg/s in case of the gyroscope and 0.195 $g$ for the accelerometer. These values hold only when maximum sensitivity settings are used. In the used case, full scale range is needed for both sensors, thus the measured deviations increase significantly. As described in Niklaus et al. (2017) these main deviation can be corrected via a linear correction function $f(x) = c_0 + c_1 \cdot x$ for each sensor axis. If applied, $c_0$ is the dominant correction term for the accelerometer in the order of 0.15 $g$ to 3 $g$. For the gyroscope, $c_1$ is dominant being of the order of 0.09% to 0.35% from the ideal value of 1.0. It is to note, that the measured offsets lie below 1% of the full scale range and thus can be neglected. For the presented trajectory the actual sensor offset at rest amount to $4.5 \pm 0.1$ deg/sec and $1.17 \pm 0.48$ $g$ as opposed to the ideal value of 1 $g$.

The free fall analysis of the measurements began immediately after the initial motion, at 54.5 s. At this time its rotation is relatively low (180.5 deg/s) and its influence on the acceleration value small. Equation 1 yields

$$a_Z = R_e \cdot \omega^2 = 0.165 \cdot \left( \frac{180.5}{360} \cdot \pi \right)^2 = 0.41 \text{ m/s}^2 \tag{2}$$



**Table 1.** Mean values of absolute rotational and acceleration data for calculating eccentricities.

| Time (s) | 59.0-59.7 | 59.9-61.0 | 61.2-61.9 |
|---|---|---|---|
| Rotation (deg/s) | 3690 | 3501 | 4098 |
| Acceleration (m/s$^2$) | 16.20 | 14.01 | 18.9 |
| Eccentricity (m) | 0.004 | 0.004 | 0.004 |
| Eccentricity (mm) | 3.91 | 3.75 | 3.70 |

Eccentricity was analyzed between 59.0 and 61.9 s, ignoring data from the two intervening ground contacts (Figure 1d). The mean acceleration values within these three intervals are used to feed Eqn. 1 and to determine the eccentricity radius, which is virtually identical for all three time intervals, equalling 0.004 m (Table 1).

### 4.1 Duration of ground contacts

Ground contacts are clearly recognizable from the measured rotational and acceleration values. Very short ground contacts last 8-15 ms, medium-length contacts 20-40 ms and lengthy contacts 50-75 ms. During the first 2-3 seconds after the rock has been set in motion, the duration of ground contacts increase very quickly to the peak values and then drops back to values of 10-30 ms, remaining at this level on flat terrain (Figure 2a). This corresponds to the intuitive understanding of larger scarring of rocks with higher kinetic energy. Remarkably, the total contact times which determine the trajectory kinematics amount only to 14%

of the total trajectory time of 21 s, or if the roll-out section after the last recorded impact is excluded, to only 6% of the total runtime.

### 4.2 Details of individual ground contacts

Individual results on absolute rotational and acceleration values during ground contacts are presented below. Here, we classified a contact as the temporal evolution between two plateaus in angular velocity. A typical contact during the acceleration phase

is shown in Figure 2b. The rotation increases with almost every - relatively short - ground contact, as at 55.24 s. This contact lasted 42 ms at a maximum acceleration of 45.6 $g$ and increased rotation from 683 to 1'087 deg/s.

The ground contact featured in Figure 2c (which lasted 28 ms, starting at 57.62 s), exhibits larger accelerations of 90.0 g while the rotation tipped from 2'921 deg/s to 2'766 deg/s indicating an opposed faced obstacle within the acceleration path. Both ground contacts shown above have clear maxima in the accelerometer data. However, some contacts with two or even

more maxima were also recorded. A relatively long ground contact occurred at 58.72 s, lasting 68 ms (Figure 2d). During this time, two main acceleration maxima were measured: 33.4 g and 30.4 g respectively. During this ground contact, rotation increased steadily from 2,758 deg/s to 3,696 deg/s.

If the angular velocity between two acceleration peaks remains constant, neither steadily rising nor falling, it indicates that two separate ground contacts occur, similar to those occurring at 63.13 and 63.15 s shown in Figure 2e. Here, the contact

times are very short (lasting 13 ms and 8 ms) and the acceleration maxima differ (138.6 g and 34.3 g). During these two





contacts, rotation increased from 4'186 deg/s to 4'387 deg/s, with a constant intermediate value to 4'334 deg/s. Interestingly, the maximum rotation of 4'709 deg/s occurred during the first contact, subsequently decreasing to the intermediate value.

Towards the end of the trajectory, the decrease in rotation occurred at much shorter time intervals than the increase on steeper terrain. A typical example thereof is presented here, a relatively short ground contact at 70.81 s, lasting 13 ms. During this time, rotation decreased from 1'831 deg/s to 1'539 deg/s, reaching a local minimum of 1'458 deg/s in between. The maximum acceleration for this ground contact was 72.5 g (Figure 2f).

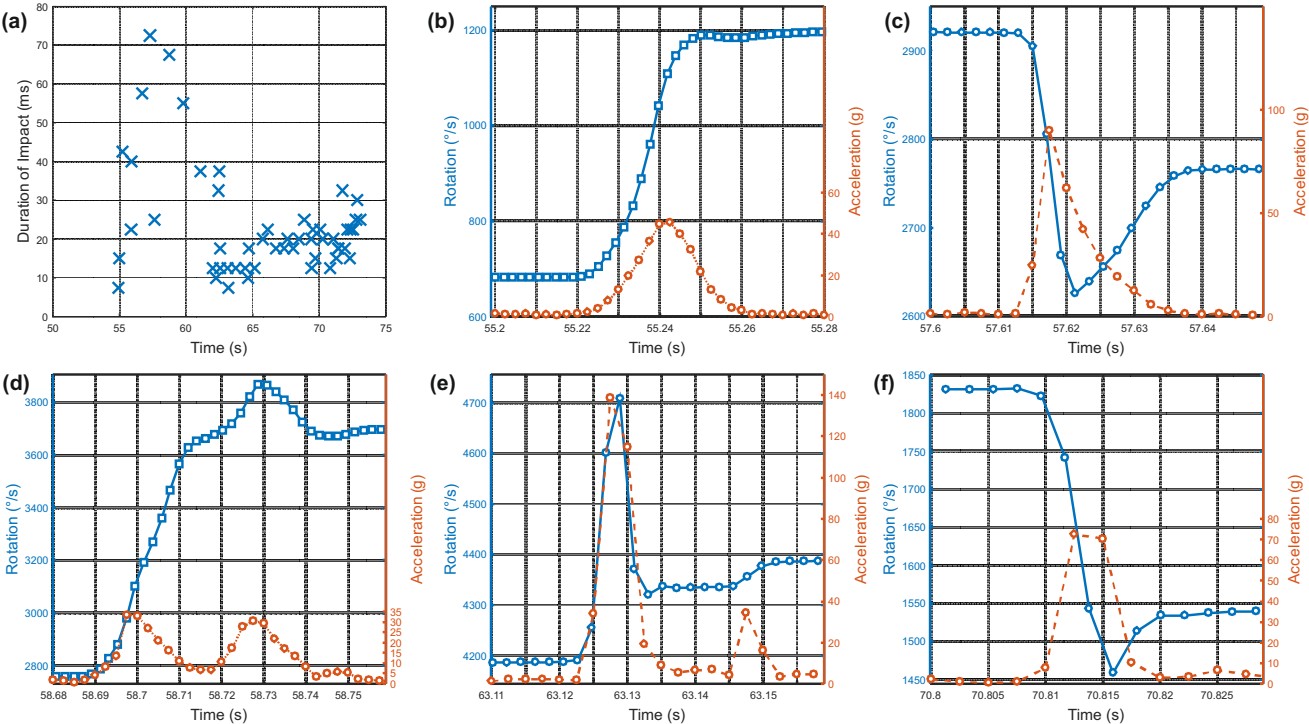

**Figure 2. (a)** Duration of ground contacts. **(b)** Absolute rotational and acceleration values for the ground contact at 55.24 s lasting 42 ms, **(c)** at 57.62 s lasting 28 ms, (d) at 58.72 s lasting 68 ms, (e) a double contact from 63.13 s onward lasting 13 ms and 8 ms (f) at 70.81 s lasting 13 ms. The error on individual measurements is smaller than the plotted marker size.

## 5   Discussion

Because the acceleration and rotation sensors exhibit very small inherent offsets, a correction is not mandatory - but feasible if desired. The constant offsets being smaller than 0.1% for the gyroscope and 0.8% for the accelerometer with respect to the full range capacity of each individual sensor undermine the high quality sensor stream. An evaluation of the sensor's centrical installation in the block indicated a very small eccentricity of 4 mm. This shows, that a careful manual placement is sufficient for accurate results.



In this experiment, ground contact duration were shown to vary considerably, ranging from a minimum of 8 ms to a maximum of 75 ms. These measurements show that longer contacts occurred on steeper terrain and shorter ones on flatter terrain. However, no precise characterization is possible yet, because the spatial data cannot be linked to the temporal data within the needed accuracy.

Temporal information on the block's flight duration between ground contacts can be used to calculate the jump height of the flight parabola (Gerber, 2015). A temporal and/or spatial link could be used to calculate the jump distance on the slope, but no such link has been established yet.

The measurements suggest very different forms of contact, both in terms of acceleration and rotation. For very short contacts of less than 10 ms, the individual measurements are not quite as reliable as the quality control purports. To measure such short
contacts, the measurement frequency would have to be increased, which is achieved by updating to **StoneNode** v1.1, which has an increased sampling rate of 1 kHz for the accelerometer and gyroscope (Caviezel et al., 2018c).

## 6 Conclusions

These measurements show that high-quality, detailed and reliable analyses of rotational and acceleration data for rocks hitting the ground are possible. The applied sensors and measurement techniques provide a logistically simple, but effective tool to
obtain kinematic data for falling rocks. The measured data provide insight to highly dynamic impact processes, but additionally raise new questions, primarily concerning the spatial relation of the rock to the surface of the terrain. For example, the rock's velocity vector at the onset of a contact relative to the slope of the surface should be known to evaluate the response of the ground material. Clearly the rock-based sensors must be combined with high-resolution, external remote sensing methods such as photogrammetry, LiDAR or RADAR to obtain the needed information. Most of these techniques are well adapted
and tuned to quasi-static conditions, that is difference mapping and/or long-term monitoring. Extending these time-of-flight measurements techniques to track a rockfall trajectory in real time fails to date due to insufficient range, resolution and/or frame rate capabilities (Horaud et al., 2016). Possible solutions are a high frame-rate, simultaneously triggered multi-camera setup and subsequent stereo-graphic reconstruction of the trajectory or a highly specified time of flight camera such as a scannerless LiDAR system capable of tracking motions as fast as 100 km/h in single reflection mode over large distances. For experimental
purposes, being interested in direct flight kinematics, these approaches might be favored over seismic signal analysis (Hibert et al., 2017; Dietze et al., 2017).

We have mainly gathered and processed temporal information from the rockfall sensors. A connection to the spatial extent of the trajectory is still missing. An approach involving the projected longitudinal profile is available, but the exact connection to the inclination of the terrain or the assignment of slopes to each ground contact is not yet possible. This information would
provide a better explanation of the general increases and decreases in rock rotation.

The sensor data are ideal to calibrate constitutive relationships which are at the kernel of the **RAMMS** rockfall software module (Caviezel et al., 2018c, b). The combination of real-terrain measurements coupled with non-smooth modelling approaches opens many new possibilities to investigate how terrain influences rock motion. Because terrain is seldom homogeneous and



rock shapes far from symmetric, in-situ measurements are needed to measure the forces at play at any given time and impact, but also for every possible rockfall trajectory. Simulated results can now be calibrated to measured data, to provide a calibration methodology for rockfall simulation codes.

One important conclusion from our measurements is that it is not possible to describe the complicated rock-soil interaction
5   process with restitution coefficients alone. Restitution coefficients describe the relationship between the incoming and outgoing velocities but provide no information concerning the decelerating forces at impact. These forces depend on the impact configuration and therefore the orientation of the rock. They act over short time periods and are impossible to average or linearize. Without a methodology to consider rock geometry, surface roughness, and soil scarring of the rock-soil interaction process is overly simplified and cannot be effectively used to make consistent runout or jump altitude forecasts.



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
