# Peer review of "Brief Communication: Measuring Rock Decelerations and Rotation Changes During Short Duration Ground Impacts"

_Natural Hazards and Earth System Sciences, 2018_

## Referee Comment (RC1) · F. Noël (Referee) · 13 Jun 2018

It's been a pleasure to read this very interesting work. Working with instrumented blocks will helps understanding the complex behaviour of falling blocks when they interact with the ground, and your work gives a good overview of this.

Specific comments:

- On p.2 line 19: "The measurements reveal the limits of field analyses", This requires a comparison, or at least a mention of what is missing with actual field analyses compared with the proposed method in the core section of the document (eg. in discus-

sion).

- The last paragraph of the conclusion section can't be written as an important conclusion of the work if its content is not shown in method/result/discussion. Coefficient of restitution (COR) is only mentioned once in the introduction, and vaguely described. Also, it is the result of the complex particle-soil interaction that happens at impact; so technically, COR might be compatible with your work if determined from the interaction of rock geometry, surface roughness... Where there might be an issue, is estimating jump altitude and runout with constant COR. But to conclude about that, it would need comparison, or at least more detail about COR. For now, I suggest the last paragraph to be modified or removed.

Here is a list of small technical corrections or improvement I suggest:

- On p.1 line 10: form -> from

- On p.3 line 10: a sphere of what material/density?

- On p.3 line 17: adding the equivalent of 4000°/s in SI (rad/s) should be considered, turn/s could be also interesting.

- On 2.2 Sensor section, the sampling rate of 400 Hz and 487.5 Hz should be mentioned (this is more a methodology characteristic than a result).

- The term "central acceleration" should be switch to "centripetal acceleration" or "radial acceleration", and/or maybe previously described, eg. "acceleration felt due to an offset with the center of mass of the block when the particle is rotating".

- On p.3 line 31: Only if there is an offset with the sensors and the center of mass... Higher acceleration values.

- On p.4 Fig. 1 d) description: Specify what are the mean acceleration values (white dots I suppose).

- On p.4 line 1: I think adding "During free fall, the rotational velocities..."

- On p.4 line 5: The last sentence could be removed because g acceleration isn't perceived by the accelerometers during free fall (so it works as well with vertical rotation).

---

## Short Comment (SC1) · 4 Jul 2018

Dear F. Noël,

thank you very much for your positive review.

We ammended the manuscript according to your suggestions. The specific comments are adressed. p.2 line 19: Foremost, we want to show how future experiments might have to be designed in order to gather data for calibration of constitutive relationship for dynamic rockfall models. This is now also expressed in the manuscript.

We agree, that more comparison with COR models is necessary to conclude in the

earlier fashion. Thus we decided to remove the last paragraph for the time being and address comparisons in future work.

The small technical corrections have been performed.

Please also note the supplement to this comment: https://www.nat-hazards-earth-syst-sci-discuss.net/nhess-2018-89/nhess-2018-89-SC1-supplement.pdf

**Supplement:**

[revised manuscript text omitted]

---

## Referee Comment (RC2) · Anonymous Referee #2 · 30 Aug 2018

This paper is dedicated to the presentation of preliminary experimental results on rock rebound using an innovative system of sensors. The general presentation of the problem tackled and the interest of the research is well presented. In addition, the sensors system, its calibration, and the assessment of its accuracy are detailed in a satisfactory manner. In a second part, experimental results obtained for a limited number of rock rebound are analyzed. These results show that the methodology proposed is relevant: it will allow a better understanding of the processes occurring during the rebound of a rock. I therefore recommend publication of this research work as a brief communication.

However, I have several minor comments / points that should be answered prior to publication. 1) Abstract : last sentence "and the basic . . .mitigation". I think that your results do not provide information on this point. It can be deleted. 2) p. 5 l. 10 : please could you better explain this point ? 3) p. 6 l. 8-9 and l. 15 : please could you clarify ? 4) conclusion – last paragraph : for sure, a more detailed and exhaustive analysis of a larger amount of results, mixed with numerical simulations, will show that it is almost not possible to obtain accurate modeling of rock propagation using restitution coefficients. But, in this paper, this is not demonstrated. I would advise to suppress this paragraph.

---

## Author Comment (AC1) · 13 Sep 2018

Andrin Caviezel and Werner Gerber

andrin.caviezel@slf.ch

Dear referee, thank you very much for your positive review.

We ammended the manuscript according to your suggestions. The specific comments are adressed. 1) Abstract: we deleted the last part of the sentence. 2) p5. l.10.: the focus on distinct impact behaviour versus rollout behaviour is highlighted and the confusing comparisong of number of impacts deleted. 3) p.6. l8-9, l.15: We rephrased the sentence slightly, such that the deeper and longer scarring behaviour of rocks with higher kinetic energy should be more clear. The change of rotational speed is rather a function of slope inclination. This observation has been added to the existing explana-

tions. p.2 line 19: Foremost, we want to show how future experiments might have to be designed in order to gather data for calibration of constitutive relationship for dynamic rockfall models. This is now also expressed in the manuscript. We agree, that more comparison with COR models is necessary to conclude in the earlier fashion. Thus we decided to remove the last paragraph for the time being and address comparisons in future work.

Best regards, Andrin Caviezel

Please also note the supplement to this comment:
https://www.nat-hazards-earth-syst-sci-discuss.net/nhess-2018-89/nhess-2018-89-AC1-supplement.pdf

---

## Author Comment (AC2) · 27 Sep 2018

(Repost from earlier response, was not checked as author response).

Dear F. Noël,

thank you very much for your positive review. We ammended the manuscript according to your suggestions. The specific comments are adressed. p.2 line 19: Foremost, we want to show how future experiments might have to be designed in order to gather data for calibration of constitutive relationship for dynamic rockfall models. This is now also expressed in the manuscript.

[Figure]

We agree, that more comparison with COR models is necessary to conclude in the earlier fashion. Thus we decided to remove the last paragraph for the time being and address comparisons in future work.

The small technical corrections have been performed

Please also note the supplement to this comment:
https://www.nat-hazards-earth-syst-sci-discuss.net/nhess-2018-89/nhess-2018-89-AC2-supplement.pdf